# Short Term Caloric Restriction and Biofeedback Enhance Psychological Wellbeing and Reduce Overweight in Healthy Women

**DOI:** 10.3390/jpm11111096

**Published:** 2021-10-26

**Authors:** Alexander Kautzky, Kathrin Heneis, Karin Stengg, Sabine Fröhlich, Alexandra Kautzky-Willer

**Affiliations:** 1Department of Psychiatry and Psychotherapy, Medical University of Vienna, 1090 Vienna, Austria; alexander.kautzky@meduniwien.ac.at; 2Department of Internal Medicine III, Division of Endocrinology and Metabolism, Medical University of Vienna, 1090 Vienna, Austria; kathrin.heneis@meduniwien.ac.at; 3Gender Institute, 3571 Gars am Kamp, Austria; 4“La Pura” Womens Health Resort, 3571 Gars am Kamp, Austria; stengg@lapura.at (K.S.); froehlich@lapura.at (S.F.)

**Keywords:** obesity, caloric restriction, biofeedback, women’s health, psychoneuroendocrinology, prevention, mental health

## Abstract

Obesity is highly prevalent, causing substantial cardiovascular and mental health morbidity. Women show increased risk for mental health disorders, that is multiplied in obesity and related to cellular and psychological stress that can be targeted by non-pharmacological interventions. A total of 43 women underwent two weeks of caloric restriction, half of which also received 7 h of individualized clinical psychological intervention including psychoeducation, mindfulness, and heart-rate-variability biofeedback. Effects on body mass index (BMI), fatty liver index (FLI), bioimpedance measures, serum parameters, perceived stress (PSS), burn-out susceptibility (burn out diagnostic inventory) and dimensional psychiatric symptom load (brief symptom inventory, BSI) were analyzed with linear mixed effects models. Caloric restriction led to a reduction in BMI, body fat and FLI, decreased serum concentrations of leptin, PSS score, BSI dimensions and global severity index (all *p* ≤ 0.0001, withstanding Bonferroni–Holm correction). Benefits of add-on biofeedback were observed for BMI reduction (*p* = 0.041). Caloric restriction was effective in ameliorating both psychological wellbeing and metabolic functions following a BMI reduction. Biofeedback boosted effects on BMI reduction and the combinative therapy may be protective against common progression to mental health and cardiovascular disorders in overweight women while comparing favorably to pharmacological interventions in terms of side-effects and acceptability.

## 1. Introduction

Overweight and obesity are becoming increasingly common and were estimated to affect respectively 40% and 10% of the global population in 2016 [1].

Overweight dramatically increases the risk for diabetes mellitus (DM) type 2 and other metabolic and cardiovascular disorders [2]. Similar to other chronic diseases such as mental health disorders, prescribing medication has been demonstrated to be deficient to reach satisfactory outcomes for many patients [3].

Many psychiatric disorders, including the most common affective and anxiety disorders, are considerably more prevalent in women [4]. Furthermore, women are known to be more exposed to stressors and morbidity than men, making them a risk-population in both metabolic and mental health that may benefit most from preventive measures [5].

Associations between obesity and mental health disorders are increasingly adopted by clinicians in both psychiatry and endocrinology [6,7,8]. Biological links range from subclinical inflammation and molecular interplay between insulin and neurotransmitters to shared unfavorable genetic, epigenetic and metabolomic risk. Excessive body fat is known to cause subclinical inflammation and hypothalamic-pituitary-adrenal axis (HPA) dysregulations are known to occur early on in obese patients [9]. The HPA, predominantly regulating the biological stress response by hormones such as cortisol, plays a key role in both endocrine and mental health [10]. However, the contribution of psychosocial factors is also evident. Suffering from core symptoms of depression such as reduced energy clearly increases the risk for adopting an unfavorable lifestyle with low levels of physical activity and reduced self-care that brings along a higher risk for obesity.

Despite effective treatment options available for both, metabolic and mental health disorders, oftentimes symptom control is hard to achieve, and disease progression cannot be stopped [11,12]. In fact, up to a third of patients with major depressive disorder (MDD) does not reach sufficient improvement in symptoms within the first months of treatment and up to 15% can be expected to reach treatment resistance [13]. Importantly, integrative treatment protocols that aim at patient education and assistance in participation have shown promising results in both endocrine and psychiatric research [14,15].

Reducing psychological and biological stress and enhancing metabolism early on may be especially effective for disease prevention. Caloric restriction (CR) has recently been advocated for normalization of insulin sensitivity and glycemic control [16,17,18]. Furthermore, positive effects on mood and cognitive performance were proposed [19,20], putatively mediated through enhancement of neuronal plasticity [21].

Similarly, biofeedback is a promising non-invasive tool to ameliorate both mental health and cardiometabolic dysfunction. Reduced heat rate variability (HRV) is a recognized feature of MDD [22], indicating impaired cardiac adaptation to new stimuli and stress [23]. Interventions aimed at increasing HRV have just recently been added to antidepressant protocols, mostly as an augmentation to cognitive behavioral psychotherapy [24,25,26].

## 2. Materials and Methods

### 2.1. Sample

Baseline characteristics and the recruitment protocol of the sample were described before [27], and details concerning all parameters included in analyses are provided in Appendix A. In short, women above 18 years of age were recruited at the VAMED gender institute (https://www.vamed.com/en/company/gender-institute/, last accessed on 20 October 2021) in cooperation with the “Gender Medicine Unit” of the Medical University of Vienna between 2019 and 2020. In total, 43 women (mean age 53.42 ± 11.7) interested in a combined psychological and diet intervention tailored for stress reduction and achieving a healthier body composition were assessed by trained clinicians and psychologists during a stay at the “La Pura Women’s Health Resort” (https://www.lapura.at/, last accessed on 20 October 2021). All women were screened clinically for eating disorders by trained healthcare staff of the “La Pura” Women’s Health Resort and all participants showed a BMI above the 25th percentile of the normal range (>20) at participation, Participants had to be not currently pregnant and screened negative for any severe or chronic illness other than obesity, hypertension, and metabolic syndrome. As described before [27], none of the women had diabetes. A total of 7% of women showed metabolic syndrome according to the definition of the world diabetes federation, requiring the presence of central adiposity together with any two of the risk factors triglycerides: ≥150 mg/dL, HDL cholesterol <50 mg/dL, blood pressure ≥130 systolic or 85 mmHg diastolic, respectively, or fasting plasma glucose ≥ 100 mg/dL [28]. None of the women had a diagnosis of liver disease. All subjects consented to the study procedures after having received detailed oral and written education. The study and all related measures were approved by the responsible Ethics Committee.

### 2.2. Treatment Groups

Women were subsequently randomly allocated to two groups, receiving either (1) caloric restriction of either F.X. Mayr or very-low calorie diet (VLCD), or (2) caloric restriction and a clinical psychological intervention (CPI) (Figure 1).

The CPI consisted of seven sessions with a trained clinical psychologist that included two individualized psycho-educative talks regarding stress prevention, two sessions of Jacobson muscle relaxation and mindfulness training, and three sessions of biofeedback. Each session lasted 50 min and sessions were appointed over a time frame of 14 days. Biofeedback sessions aimed at controlling and improving biological functions, such as breathing rhythm, pulse, and blood pressure, with a focus on HRV. Serving as a proxy marker of emotional regulation that can be measured and displayed in real-time, HRV is directly targeted during biofeedback training by relaxation and breathing exercises.

The group without CPI, receiving only caloric restriction, was invited to a 50-min lecture on stress prevention held by a trained clinical psychologist and was free to choose from the repertoire of wellness and health-focused activities provided by the “La Pura” Women’s Health Resort at the time of stay. Unrelated to the study protocol, these offers included sport programs, balneotherapy and massages.

VLCD restricts calorie intake to 630–700 kcal per day split upon three meals. Following the protocol of Taylor et al. [17], carbohydrates, fat and proteins were reduced in equal measure to normalize glycemic control and enhance long-term weight loss (20% fat, 34% protein, 46% carbohydrates).

F.X. Mayr diet similarly applies calorie restriction to 700–800 kcal per day, however, is not primarily designed to reduce weight [29]. Instead, F.X. Mayr diet aims at stimulating gastrointestinal peristaltic and the production of bile acids and reducing daily insulin peaks. This is achieved by dietary rules such as prolonged chewing at a slower pace, food combining [30] and no food intake later than 7 p.m., together with calorie reduction and daily ingestion of isotonic magnesium sulfate solution.

All women were asked to document physical activity during their stay at the “La Pura” Women’s Health Resort to allow calculation of the metabolic equivalent of task (MET) [31]. For each subject, physical activity during the study period was estimated by summed METs.

### 2.3. Measures

Anthropomorphic parameters and serum parameters were measured at baseline and after the study period of 14 days. Body mass index (BMI), waist circumference, body fat and lean mass assessed by bioelectric impedance analysis, as well as the waist-to-height-ratio (WHtR) were registered.

Serum parameters included cholesterol (HDL and total), triglycerides, sex hormones (testosterone, estradiol, follicle stimulating hormone (FSH) and luteinizing hormone (LH)), adipokines (leptin, secretagogin, resistin and adiponectin), alkaline phosphatase (AP), gamma glutamyltransferase (GGT), the pro-form of brain natriuretic peptide (proBNP), immuno-inflammatory markers c-reactive protein (CRP) and interleukin-6, and serum as well as saliva cortisol. Liver function and risk for non-alcoholic fatty liver disease (NAFLD) was primarily assessed by fatty liver index (FLI) that was calculated from triglycerides and GGT [31].

For assessment of serum brain-derived neurotrophic factor (BDNF) levels, the BioVendor^®^ human BDNF ELISA kit was applied (BioVendor, Brno, Czech Republic). In more detail, plasma was collected at 08:00 in the morning after overnight fasting and probes were frozen at −20 °C before processing. All preparations, assay procedures and BDNF calculation were performed according to the product manual.

In addition, all subjects were analyzed by bio-feedback assessment using the NeXus 10 from MindMedia (6049 CD Herten, Netherlands, www.mindmedia.com/de/produkte/nexus-10-mkii/, last accessed on 20 October 2021). Biofeedback parameters were assessed in a resting state at baseline (20 min) as well as during stress exercises. These consisted of three consecutive stress tests (Stroop test [32], serial sevens [33], thinking of recent personal stressful experience) of three minutes each that were separated respectively by one minute of relaxation paradigms (looking at a relaxing image). HRV at baseline and under stress and systolic and diastolic blood pressure were used for analyses.

Psychometric assessment was also performed at baseline and at Day 14 of the study period. Thereby, the brief symptom inventory (BSI) was applied. The BSI provides a symptom score for psychiatric symptom dimensions as well as a global severity index (GSI) [34]. For assessment of burnout, the recently developed “burnout dimensions inventory” (BODI) was applied [35], composed of self-rating scores for the subitems of work, family and self-oriented stress as well as four composite items describing reduced resilience, reduced ability of distancing, depression and dysfunctional decompensation. Stress was also evaluated by the perceived stress scale [36].

A full characterization of all parameters by study visit and group can be found in Appendix A.

### 2.4. Statistical Analyses

Linear mixed models were computed with the statistical software “R” (https://www.r-project.org/, last accessed on 20 October 2021) applying the “nlme” package [37]. Each model was computed with the interaction term group * visit, therefore including the two-way interaction between treatment group (binomial) and visit (binomial; Day 1 and Day 14) as well as respective main fixed effects and patient identifier as random effect. Deviance tables were generated with the ANOVA function for “nlme” and respective F- and *p*-values are reported.

Results were corrected by Bonferroni–Holm test. Consequently, a *p*-value below a threshold of 0.0005 was required to withstand correction. Uncorrected *p*-values are reported and findings that did not withstand correction are referred to as trends.

## 3. Results

### 3.1. Anthropomorphic Parameters

Groups did not differ in glucose levels, HbA1c and functional parameters for insulin sensitivity (Matsuda Index) or resistance (HOMA-IR) at baseline. A total of 67.4% of women scored above a BMI of 25, with 46.5% and 20.9% of women classified as overweight and obese, respectively. Further, 42.9% of obese women scored a BMI above 40, indicating Obesity Grade III. Baseline BMI ranged from 20 to 42.9 kg/m^2^.

BMI was significantly lower after the study period (F = 89.3, *p* < 0.0001). An interaction effect between time and CPI was observed, that did not withstand correction for multiple testing (F = 4.4, *p* = 0.041). Average BMI reduction was 0.67 kg/m^2^ in women without CPI and 1.1 kg/m^2^ in the women with add-on CPI. Physical activity during the study period was higher in women without CPI (F = 4.82, *p* = 0.036).

Lower scores for WHtR (F = 43.5, *p* < 0.0001) and body fat (F = 49.8, *p* < 0.0001) were observed after the study period and a trend was observed for decreased lean mass (F = 5.3, *p* = 0.031).

At baseline, 23.3% of patients showed a high risk for suffering from NAFLD, as indicated by a FLI > 60. During the study period, the FLI declined in both groups (F = 18, *p* = 0.0001), and 14% of patients scored a FLI > 60 after two weeks of CR.

Please refer also to (Table 1, Section A) and Figure 2.

### 3.2. Serum and Biofeedback Parameters

Leptin (F = 56.4, *p* < 0.0001) and trendwise also adiponectin (F = 5, *p* = 0.032) and secretagogin (F = 7.1, *p* = 0.012) declined in all groups. Regarding sex hormones, an increase was observed for FSH (F = 27.1, *p* < 0.0001) as well a trend of increased testosterone (F = 4.5, *p* = 0.04).

Concerning cholesterol, strongest reductions were seen in HDL (F = 25.3, *p* < 0.0001) while a trend was observed for total cholesterol (F = 7.1, *p* = 0.011). A reduction was also found in AP (F = 33.6, *p* < 0.0001) and GGT (F = 16.7, *p* = 0.0002) and on trend level for proBNP (F = 5.31, *p* = 0.027). No effects were found for BDNF, interleukin-6 and cortisol. Among biofeedback parameters, only the systolic blood pressure decreased on trend level (F = 12.7, *p* = 0.001).

Please refer also to Table 1, Section B. For graphical representations please see Figure 2 and Figure 3.

### 3.3. Psychometric Parameters

At baseline, moderate self-perceived stress (PSSI ≥ 14) was present in 67% of women. At day 14, 54.5% of women showed low self-perceived stress ((PSSI < 14), leading to a significant reduction of PSS total scores across all groups (F = 16.1, *p* = 0.0003). Additionally, a trendwise reduction in BODI Item 3, reduced ability of distancing, was observed (F = 10.2, *p* = 0.003). Self-rating scores for as self-oriented stress was over time on trend level (F = 5.7, *p* = 0.023).

A decline in GSI (F = 52.8, *p* < 0.0001) was observed, with clinically relevant scores (GSI ≥ 63) in 32.5% of women at baseline and 4.7% of women at Day 14. Similarly, BSI subscores for depression (F = 25.1, *p* < 0.0001), anxiety (F = 30, *p* < 0.0001), paranoia (F = 35.7, *p* < 0.0001), psychoticism (F = 22.5, *p* < 0.0001) compulsion (F = 25.4, *p* < 0.0001) and aggression (F = 21.1, *p* = 0.0001) declined, on trend level also somatization (F = 6.9, *p* = 0.012).

Please refer also to Table 1, Section C. For graphical representations please see (Figure 4).

## 4. Discussion

Two weeks of CR showed favorable results concerning decrease in body fat and waist circumference, indicating reduced visceral fat that may represent metabolic adaptation protective of a broad range of diseases such as DM 2, cardiovascular and neurodegenerative events and oncologic disorders [38].

Other common findings after CR include changes in lipids, sex hormones and less consistently inflammation markers such as CRP [39,40]. Here, cholesterol parameters declined over the study period while CRP, cortisol and interleukin-6 did not show relevant changes. More surprisingly, we observed an increase in testosterone and FSH in all groups, contrasting earlier reports [41,42]. Considering the small sample size together with the short duration of the treatment period, rather favorable baseline values and the majority (79%) of women being in peri- or post-menopause, changes in sex hormones and immunological markers should be interpreted cautiously.

Considering that obesity is a major risk factor for developing NAFLD, the general reduction of FLI after two weeks of CR is a promising finding. However, our data did not support significant effects of two weeks of CR and CPI on IL-6, which was suggested an important risk marker for NAFLD in obesity [43]. On the other hand, a mean reduction in IL-6 of 21% (*p* > 0.05) was observed in those 23.3% of women who showed a high risk for presence of NAFLD. Consequently, protective effects may only be seen in a sample with more unfavorable clinical characteristics such as a high FLI or after longer CR protocols, which was suggested by some earlier studies [39]. Along these lines, despite a reported good specificity for FLI values < 30 and sensitivity for values > 60 for respective absence and presence of NAFLD, we cannot provide a definitive diagnosis since no further assessments such as sonography were performed [44]. Here, CR led to improvement of mental health, reflected by a decline in the BSI as well as perceived stress. Despite little data on CR in psychiatric treatment, a potential role of CR in affective disorders is a long standing but also timely finding [19,45,46,47]. Antidepressant qualities may be triggered by an adaptation to reduced glucose availability [20,48], leading to reduced radical oxygen species, increased lysosomal autophagy and changes in adipokines and mitochondrial functioning [49,50].

CR caused an average decrease of 33% in leptin concentrations. Leptin was shown to have direct neuroactive properties via leptin-receptors expressed in several brain regions [51,52]. Additionally, leptin extenuates overactivation of the HPA axis in rodent models. On the other hand, leptin also causes oxidative stress and most recent studies rather reported an increase in leptin concentration in depressed patients [53,54]. Despite leptin being reduced by CR, both putatively stimulate neuroplasticity via similar pathways [55,56]. In rat models, CR but also leptin injections were shown to directly enhance neurogenesis and cell survival in the hippocampus, a well-established core region of structural and functional impairment in depression [57,58]. A possible solution may be leptin resistance in overweight patients and consequently a dissociation of leptin concentration from proper functioning [59]. The reduction of weight and leptin with concomitant amelioration of mental health in this study further strengthens a systemic role of leptin.

Consequently, CR shows overlap with the therapeutic mechanisms of commonly prescribed antidepressants and electro-convulsant therapy [60]. While we did not observe significant changes in BDNF levels, a lack of quick changes but rather increases after continuous treatment were also reported for non-rapid-acting antidepressants. Further, baseline differences between groups, with lower BDNF observed in the CPI groups compared to women receiving only dietary interventions (F = 4.65, *p* = 0.038), and a wide range of BDNF values with extreme values may hinder reliable interpretation.

CR findings in mental health are promising as most psychopharmacological agents bear a risk of weight gain and potentially deflect glucose metabolism to diabetogenic states [61]. Thus, dietary treatment options may be particularly helpful in overweight and obese patients, especially as early preventive measures. Good motivation for and adherence to self-administered CR regimens were reported at least in diabetic patients [62], addressing another decisive challenge in treatment of mental health disorders [14,15].

Augmentative effects of the add-on HRV-biofeedback and clinical-psychological intervention were less pronounced. While not reaching statistical significance, the recipients of CPI showed a greater mean decline over the study period for several stress related parameters such as BODI dysfunctional compensation (34.4% vs. 25.7%) as well as PSS score (27.7% vs. 12.3%). Interestingly, no changes in HRV could be detected between women having received CPI and those who did not. Previous integration of HRV biofeedback in MDD and anxiety patients mostly reported an increase in some functional parameters related to HRV [24]. However, the intervention applied here was considerably shorter compared to other studies on HRV biofeedback that usually implemented a training time of 6–10 weeks [25,26,63,64]. Similar to our result, a short intervention of just two weeks of biofeedback showed changes in HRV only in depressed patients but not healthy subjects, irrespective of mood changes [65]. Considering that all groups showed improvements in mental health dimensions, effects of CPI may be disguised by CR. Effects of CPI could also be influenced by between-group differences in physical activity that is beneficial for both metabolic and mental health. Considering that women without CPI had more time to spend on other activities, it may not be surprising that average physical activity was roughly 40% higher in these groups. In synopsis, the current results suggest a small add-on advantage of two weeks of CPI for BMI reduction.

Despite the Bonferroni–Holm correction, we cannot rule out false positive findings due to the small patient count and exploratory nature of the study. Results are reported in healthy women that mostly did not show clinically relevant deflections in either metabolic or psychometric parameters but still differed considerably in anthropo- and psycho-metric parameters, resulting in a high degree of heterogeneity. As a result, we were not able to further stratify the groups by implementing pre- or post-menopausal state in the analyses. Psychometric assessment was performed by a trained clinical psychologist; however, no psychiatric diagnostic interview was conducted. Further, while eating behavior was guided by the applied CR interventions, it was not formally assessed clinically or by dietary protocols. Considering that some dysfunctional eating behaviors were linked to symptoms of anxiety and depression as well as personality factors in the light of obesity [66], it may be important to control for these characteristics in future studies. Finally, women staying at the “La Pura” Women’s Health Resort can be expected to show favorable socioeconomic status and therefore may not be representative of the general female population and general psychological effects mediated by the stay independently of the interventions must be assumed. This limitation is emphasized by the high education level of 87% of women reporting post-secondary education observed in this sample.

In synopsis, short-term CR and biofeedback in healthy women proved effective for reduction of psychological stress, mental health symptom load, as well as metabolic parameters. Differences between CR regimens and the benefit of add-on HRV biofeedback as well as long term changes in neurotrophic factors such as BDNF call for further investigation, especially in relevant patient groups with more severe deflections in metabolic and mental health. Nevertheless, our results advocate CR as a preventive tool for both metabolic and mental health disorders that compares favorably to pharmacological treatment in terms of relevant side effects and can easily be self-administered with little professional support [26,62]. Combined with reported high interest and good acceptability in the patient and at-risk groups, these non-pharmacological treatment options may substantially enrich the canon of preventive and augmentative treatment options in a broad spectrum of mental health and metabolic disorders, considering that obesity may be one the most critical moderating factor that can be reversed by patients themselves.

## Figures and Tables

**Figure 1 jpm-11-01096-f001:**
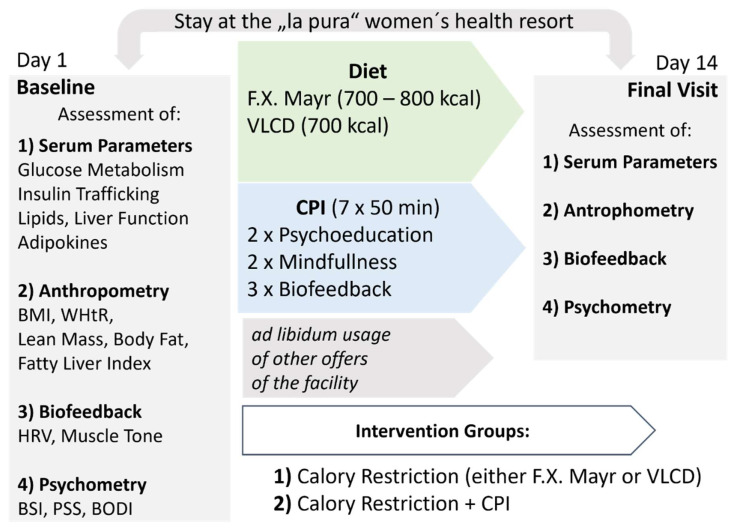
Timeline and rationale of the study design. Additional offers provided by the facility included sport programs, balneotherapy and massages, and were not restricted by the study protocol. Abbreviations: CPI = clinical-psychological intervention, VLCD = very-low-calorie diet, BMI = body mass index, BSI = brief symptom inventory, PSS = perceives stress scale, BODI = burnout dimensions inventory.

**Figure 2 jpm-11-01096-f002:**
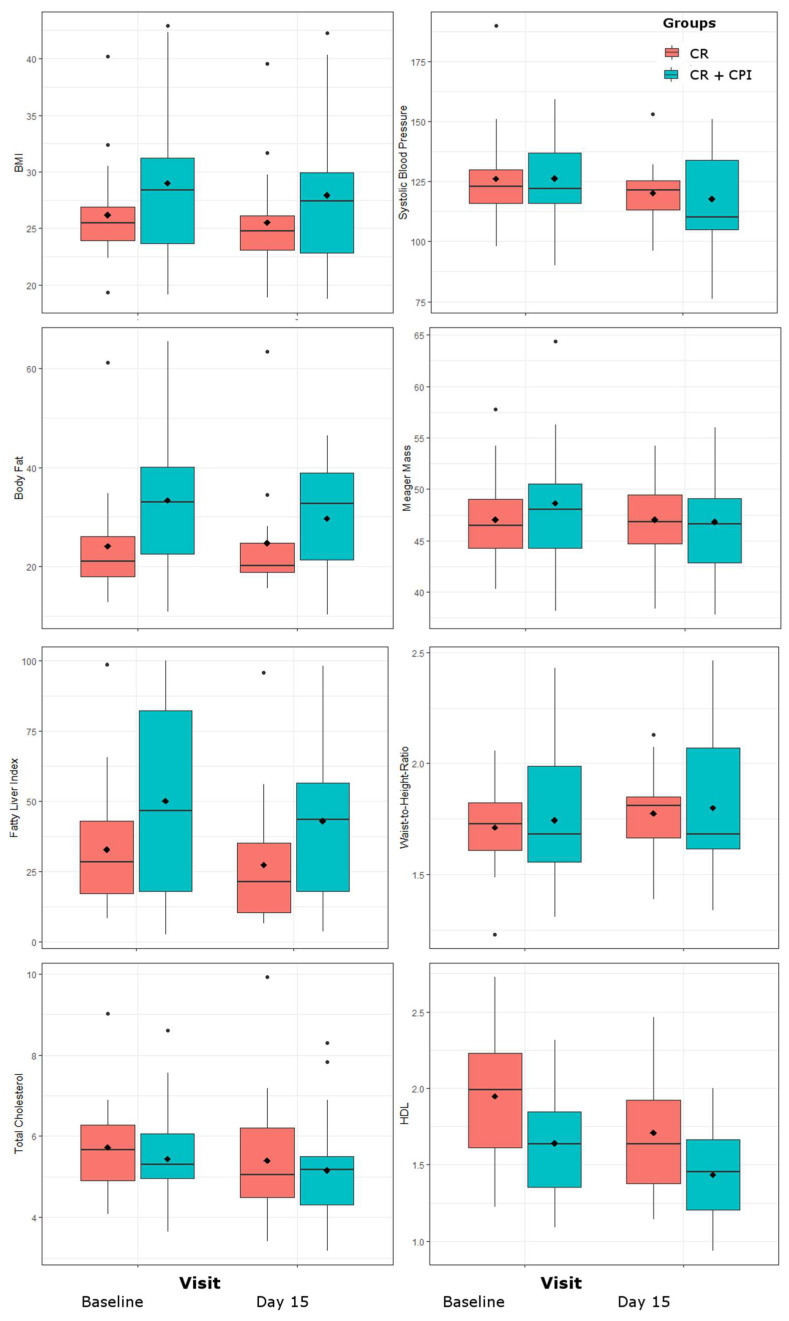
Changes in anthropometric parameters body mass index (BMI), waist-to-height-ratio, fatty liver index, as well as total and HDL cholesterol and systolic blood pressure over the study period of 14 days. Values are grouped by treatment groups of caloric restriction (CR) alone and CR together with clinical psychological intervention (CPI).

**Figure 3 jpm-11-01096-f003:**
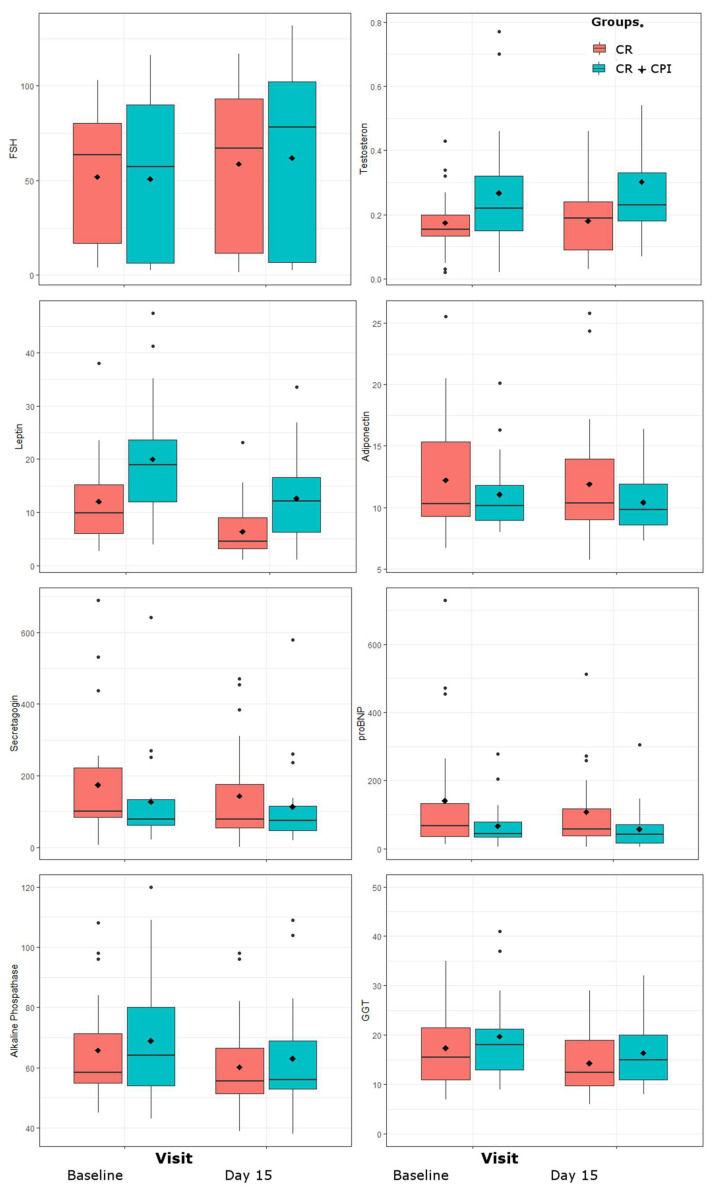
Changes in adipokines, sex hormones follicle-stimulating hormone (FSH) and testosterone, alkaline phosphatase, gamma-glutamyl-transferase (GGT) and pro-brain-natriuretic-peptide (proBNP) over the study period of 14 days. Values are grouped by treatment groups of caloric restriction (CR) alone and CR together with clinical psychological intervention (CPI).

**Figure 4 jpm-11-01096-f004:**
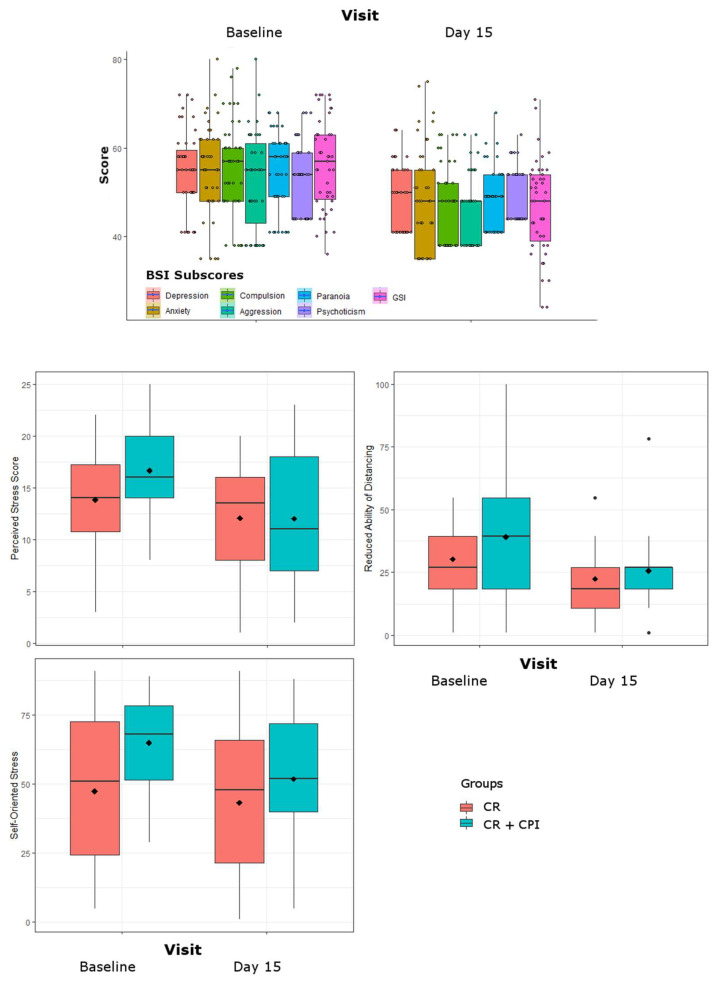
Changes in psychometric parameters over the study period of 14 days. Dimensional values of the brief symptom inventory (BSI) are provided for the two study timepoints as well as the global severity index (GSI). Results for the BODI Item 3 (reduced ability of distancing), the BODI self-rating of self-oriented stress and the perceived stress scale total score are grouped by treatment groups of caloric restriction (CR) alone and CR together with clinical psychological intervention (CPI).

**Table 1 jpm-11-01096-t001:** Mixed model results, grouped by types of outcome parameters. For each outcome of interest, models with main effects of visit (baseline and after two weeks) and group (caloric restriction alone, and in combination with clinical-psychological intervention—CPI), as well as their interaction effect were calculated. Only significant effects are listed. Numerator and denominator degrees of freedom (numDF, denDF), F and *p*-values are reported. Effects of visit indicate similar results in both treatment groups and thus caloric restriction. Uncorrected results are displayed. Interaction effects are indicated by *.

Outcome	Predictor	numDF	denDF	F-Value	*p*-Value
(A) Anthropometric Parameters					
Body Mass Index	Visit	1	39	87.990	<0.0001
	Visit * CPI	1	41	4.4	0.041
Waist-to-Height Ratio	Visit	1	40	43.5	<0.0001
Waist Circumference	Visit	1	38	36.6	<0.0001
Body Fat	Visit	1	21	49.8	<0.0001
Lean Mass	Visit	1	22	5.3	0.031
Fatty Liver Index	Visit	1	39	18	0.0001
(B) Serum and Biofeedback Parameters					
Total Cholesterol	Visit	1	40	7.1	0.011
HDL	Visit	1	40	25.3	<0.0001
Testosterone	Visit	1	39	4.5	0.04
FSH	Visit	1	39	27.1	<0.0001
AP	Visit	1	39	33.6	<0.0001
Gamma GT	Visit	1	39	16.7	0.0002
proBNP	Visit	1	39	5.5	0.024
Adiponectin	Visit	1	38	5	0.032
Secretagogin	Visit	1	34	7.1	0.012
Leptin	Visit	1	38	56.4	<0.0001
Systolic Blood Pressure	Visit	1	40	12.7	0.001
(C) Psychometric Parameters					
PSS	Visit	1	31	16.1	0.0003
BODI 3	Visit	1	29	10.2	0.003
GSI	Visit	1	41	52.8	<0.0001
Interpersonal Sensitivity	Visit	1	41	24.5	<0.0001
Somatization	Visit	1	41	6.9	0.012
Depression	Visit	1	41	25.1	<0.0001
Anxiety	Visit	1	41	30	<0.0001
Aggression	Visit	1	41	21.1	0.0001
Compulsion	Visit	1	41	25.4	<0.0001
Paranoia	Visit	1	41	35.7	<0.0001
Psychoticism	Visit	1	41	22.5	<0.0001
Stress Self-Oriented (BODI Self-Rating)	Visit	1	31	5.7	0.023

## Data Availability

Data are available from the corresponding author upon reasonable request.

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
