# Peer review of "Short Term Caloric Restriction and Biofeedback Enhance Psychological Wellbeing and Reduce Overweight in Healthy Women"

_jpm, 2021, doi:10.3390/jpm11111096_

Round 1
Reviewer 1 Report
In this manuscript Kautzky and colleagues describe a study conducted in women motivated to lose stress and weight, who were subjected to caloric restriction and/or clinical-psychologic intervention over a period of 14 days in a controlled environment. The authors assessed many physiological and psychological parameters and observed significant improvements over the study period. These results support the authors’ premise that obesity and diabetes-related pathophysiological mechanisms negatively impact mental health, and that strategies aiming at improving metabolic control also improve mental health.
Overall the study is well-conducted and the manuscript is well-written. I would only like to pose some minor comments:
- The first statement of the Introduction is perhaps exaggerated - all regions of the world?
- Should major depression be referred to as MDD? Shouldn’t it be MD?
- Please double-check if “correlates” is the adequate term in line 62
- Do antidepressant protocols really include interventions aimed at increasing HRV? Or are these protocols aimed at improving the patients’ emotional status/well-being, which translates in an increase in HRV?
- There are several elements lacking in this manuscript as well in the paper by the same authors (ref 27) regarding the characterization of the study sample, namely: (1) have the authors characterized the eating behavior of the participants (some behaviors are related to anxiety/depression and increase metabolic risk); (2) can the authors provide any information on the social behavior of participants (socioeconomic status, income, level of education, etc.)?
- Also, there are some odd statements regarding the participants, namely: (1) none of the women had diabetes but 7% had metabolic syndrome – do the authors mean type 1 diabetes? If they mean type 2, then the sentence does not make sense; (2) as per section 3.1. several participants had a baseline BMI of around 20 (lower limit of the acceptable range), however the authors claim that the women who participated in this study were interested in reducing stress AND weight – does this mean that these participants had eating disorders? Or in alternative would it be more correct to write that these women were interested in reducing stress AND/OR weight? Please revise.
- Which biological functions are the authors referring to in line 96?
- Please describe the “wellness and health-focused activities” provided by the facility in line 100.
- What other offers (figure 1) did the facility provide which could be used ad libitum?
- On line 109 do the authors mean “slow chewing” or “prolonged chewing”?
- Please revise figure 1 – insulin “functioning” is not a correct term; replace “waist-height-ratio” with an acronym; replace “bioimpedance analysis” with the names of the parameters;
- The term “meager mass” is not correct; it should be “lean mass”
- Why not assess LDL alongside HDL and total cholesterol?
- Was cortisol really assessed on the sputum? Or was it assessed in saliva? Typically cortisol is assessed in saliva and the presence of sputum in a sample is a criterion for rejecting that sample. Please revise.
- The term “paradigms” (line 141) is not the most adequate; shouldn’t it be “tests”?
- Add “test” after “Bonferroni-Holm” on line 163.
Reviewer 2 Report
Authors should be congratulated for approaching an original topic.
Some suggestions that authors are advised to follow:
Authors are requested to present for each variable studied in their investigation basal values as well as values after the intervention so that readers can appreciate the clinical effective difference not being sufficient referring to another study. Indeed, the appreciation of values in Figures is a little bit approximative.
Dealing with a 2X2 model of ANOVA was considered the interaction treatment group*visit?
Did authors use FLI and liver enzymes to evaluate liver function for detecting the NAFLD (strictly linked to obesity) presence ?
Why did they not mention the presence of this liver disease, at the light that there are some people with metabolic syndrome (7%) and nearly 40% with an obesity of III grade ?
How authors explain after intervention the lack of reduction of the main inflammatory cytokine, i.e., IL-6 that links obesity, mainly visceral obesity, and NAFLD?
This cytokine is expression of the so-called chronic low-grade inflammation, central mechanism of obesity-related NAFLD, although not the unique, as evident in J. Clin. Med. 2020, 9(1), 15; https://doi.org/10.3390/jcm9010015
Round 2
Reviewer 2 Report
Authors improved their manuscript according to comments